# G$_{i/o}$ protein-coupled receptors in dopamine neurons inhibit the sodium leak channel NALCN

Fabian Philippart, Zayd M Khaliq*

Cellular Neurophysiology Unit, National Institute of Neurological Disorders and Stroke, National Institutes of Health, Maryland, United States

**Abstract** Dopamine (D2) receptors provide autoinhibitory feedback onto dopamine neurons through well-known interactions with voltage-gated calcium channels and G protein-coupled inwardly-rectifying potassium (GIRK) channels. Here, we reveal a third major effector involved in D2R modulation of dopaminergic neurons - the sodium leak channel, NALCN. We found that activation of D2 receptors robustly inhibits isolated sodium leak currents in wild-type mice but not in NALCN conditional knockout mice. Intracellular GDP-βS abolished the inhibition, indicating a G protein-dependent signaling mechanism. The application of dopamine reliably slowed pacemaking even when GIRK channels were pharmacologically blocked. Furthermore, while spontaneous activity was observed in nearly all dopaminergic neurons in wild-type mice, neurons from NALCN knockouts were mainly silent. Both observations demonstrate the critical importance of NALCN for pacemaking in dopaminergic neurons. Finally, we show that GABA-B receptor activation also produces inhibition of NALCN-mediated currents. Therefore, we identify NALCN as a core effector of inhibitory G protein-coupled receptors.
DOI: https://doi.org/10.7554/eLife.40984.001

*For correspondence:
zayd.khaliq@nih.gov

Competing interests: The authors declare that no competing interests exist.

## Introduction

Dopamine (D2R) receptors are G$_{i/o}$ protein-coupled receptors that are expressed widely throughout the brain to control a range of behaviors including locomotion, motivation, action selection, and appetitive reward-seeking (*Gerfen and Surmeier, 2011*; *Tritsch and Sabatini, 2012*). Drugs that target D2 receptors have been used for decades as therapies against schizophrenia, but also for bipolar disorder, obsessive-compulsive disorder, Huntington's disease and Parkinson's disease. A critical step in improving therapeutic approaches that involve these and other G$_{i/o}$ protein-coupled receptors is to obtain a more complete knowledge of their primary effectors.

An extensive literature has been devoted to understanding D2 receptor modulation of midbrain dopamine neurons. D2-receptors provide feedback autoinhibition of dopaminergic neurons (*Ford, 2014*). Drugs of abuse such as cocaine increase dopamine levels which then inhibits the activity of dopamine neurons (*Einhorn et al., 1988*). Genetic ablation of D2-autoreceptors leads to hyperactivity and enhanced locomotor responses to cocaine in mice (*Anzalone et al., 2012*; *Bello et al., 2011*).

At the cellular level, most studies examining D2-autoreceptor modulation focus primarily on two ion channel effectors: voltage-gated calcium channels and G protein-coupled inwardly-rectifying potassium (GIRK) channels (*Beaulieu and Gainetdinov, 2011*). In midbrain dopamine neurons, somatodendritic D2 receptors autoregulate excitability through inhibition of high-threshold voltage-gated calcium channels (*Cardozo and Bean, 1995*). In addition, D2 receptors couple to GIRK channels to hyperpolarize cells and inhibit tonic firing (*Aghajanian and Bunney, 1977*; *Beckstead et al., 2004*; *Lacey et al., 1987*). Genetic ablation of the GIRK2 subunit alone abolishes all GIRK-mediated

currents in dopaminergic neurons, resulting in a substantial reduction in both D2 and GABA-B receptor-mediated currents (*Beckstead et al., 2004*; *Cruz et al., 2004*; *McCall et al., 2017*). Interestingly, a non-GIRK component remains in GIRK2 knockout mice and has yet to be identified (*Cruz et al., 2004*; *McCall et al., 2017*). A potential candidate may be Kv1 channels that are recruited by D2 receptors to inhibit dopamine release from axon terminals (*Fulton et al., 2011*). However, examination of somatic firing in dopamine cells has shown that block of somatodendritic Kv1 channels has little effect on the rate of firing (*Khaliq and Bean, 2008*), suggesting that somatic Kv1 channels are unlikely to account for this current. Considering this evidence, the modulated current remaining in GIRK2 knockouts may be a protein class separate from potassium conductances. In this study, we investigate whether activation of $G_{i/o}$ protein-coupled receptors results in modulation of the resting sodium leak conductance.

NALCN is a non-selective sodium leak channel that is expressed widely in neurons throughout the brain (*Ren, 2011*). Mutations in NALCN leads to dysfunction of brainstem respiratory systems in mice (*Lu et al., 2007*; *Yeh et al., 2017*), disrupted circadian rhythms in flies (*Flourakis et al., 2015*), motor dysfunction in *C. elegans* (*Gao et al., 2015*), and cognitive defects in humans (*Bend et al., 2016*; *Chong et al., 2015*; *Lozic et al., 2016*). At the cellular level, NALCN-mediated currents depolarize the resting membrane potential in hippocampal neurons (*Lu et al., 2007*) and drive spontaneous firing in GABAergic neurons of the substantia nigra pars reticulata (*Lutas et al., 2016*). In addition to generating a sodium leak current, NALCN channels are activated by tachykinin- and neurotensin-receptors through a G protein-independent, tyrosine kinase-dependent mechanism (*Lu et al., 2009*). By contrast, the calcium sensor receptor (CaSR) regulates NALCN in a G-protein dependent manner (*Lu et al., 2010*). Interestingly, behavioral experiments in *C. elegans* suggest that the nematode dopamine receptor, Dop-3, modulates ortholog $Na^+$ leak channels, NCA-1 and NCA-2 (*Topalidou et al., 2017*). However, direct electrophysiological evidence for modulation is lacking and results may differ in mammalian neurons. Therefore, whether NALCN-mediated currents are regulated by $G_{i/o}$-coupled receptors such as the D2 receptor has yet to be examined.

Here, we show that NALCN-mediated sodium leak currents are strongly inhibited by activation of D2 receptors. Moreover, in the presence of GIRK channel blockers, the D2 receptor inhibition of NALCN alone decreases firing rate in dopamine neurons. In addition, this inhibition was observed following GABA-B activation. Thus, our data identify the NALCN channel as a core effector of $G_{i/o}$ protein-coupled receptors that functions along with GIRK channels to modulate neuronal firing.

## Results

### Determining the contribution of NALCN to sodium leak currents in SNc dopamine neurons

We set out to test the involvement of NALCN channels in $G_{i/o}$ protein-coupled receptor modulation of midbrain dopamine neurons. Previous work has demonstrated the expression of NALCN channels in primary dopamine neuron cultures from postnatal mice (*Lu et al., 2009*). To test the role of NALCN channels in excitability of dopaminergic neurons from adult animals, we generated mice in which the NALCN allele was flanked by lox-P sites (*Nalcn^flox*). These mice were then bred to DAT-Cre mice, resulting in offspring (*Nalcn^{flox/flox};Slc6a3^{Cre}*) that lack expression of NALCN in dopaminergic neurons.

To determine the contribution of NALCN to background leak currents in dopamine neurons in coronal brain slices from adult mice, we first isolated sodium leak currents using a $Cs^+$-based intracellular recording solution ($CsMeSO_3$) to block potassium conductances. In addition, tetrodotoxin (1 μM), CsCl (3 mM), and apamin (300 nM) were included in the extracellular solution to block voltage-gated sodium channels, hyperpolarization-gated ($I_h$) channels and small-conductance calcium-activated (SK) channels (*Figure 1*). Finally, to block resting sodium leak currents, we replaced $Na^+$ ions in the extracellular bath with a large impermeant ion, N-methyl-D-glucamine (NMDG). As shown in *Figure 1B,C*, sodium replacement with NMDG resulted in a reduction of the average holding current ($V_{hold} = -70$ mV) from $-40.7 \pm 10.18$ to $-13.75 \pm 7.39$ pA (n = 8; p = 0.0078, Wilcoxon). Consistent with previous work in dopamine neurons (*Khaliq and Bean, 2010*), these results demonstrate the presence of a small amplitude resting sodium leak conductance.

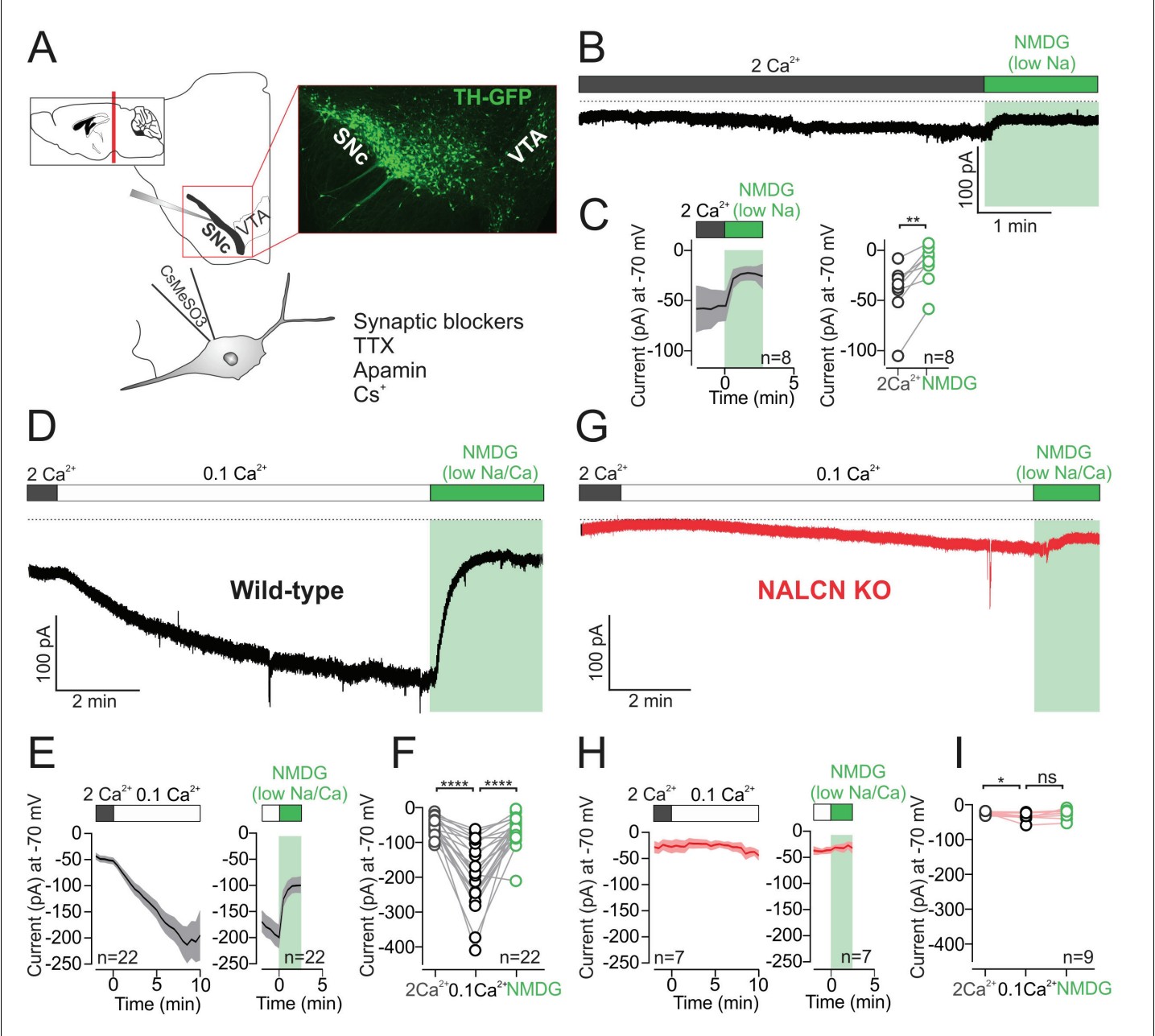

**Figure 1.** Low external calcium potentiates NALCN current in SNc dopamine neurons from adult mice. (**A**) *Top*, schematic of a coronal section (left) and confocal image of coronal section from TH-GFP mouse. *Bottom*, schematic of recording solutions, with cesium based internal and external sodium leak isolation cocktail. (**B**) Example trace of isolated sodium current recorded at −70 mV from wild-type mouse SNc dopamine neuron before and after Na$^+$ replacement with NMDG. (**C**) *Left*, Average time course of isolated sodium current before and after replacement of external Na$^+$ by NMDG. *Right*, summary plot of isolated sodium current (pA) recorded at −70 mV in control (*gray*) and in NMDG solution (*green*). (**D**) Example trace of isolated sodium leak current showing potentiation of the sodium leak in 0.1 mM Ca$^{2+}$ solution and the block induced by Na$^+$ substitution by NMDG. (**E**) Time course of the averaged 0.1 mM Ca$^{2+}$ mediated current (*left*) and the block of this current after replacement of Na$^+$ by NMDG (*right*). (**F**) Summary plot of isolated sodium current amplitude at −70 mV in control (*gray*), 0.1 mM Ca$^{2+}$ (*black*) and after Na$^+$ substitution by NMDG (*green*). (**G**) Example trace of isolated sodium leak current in NALCN cKO mouse. Note, low external Ca$^{2+}$ and Na$^+$ substitution with NMDG has relatively little effect on holding current. (**H**) Same as in E but in NALCN KO mice. (**I**) Summary plot of isolated sodium current in control (*open symbols, gray*), 0.1 mM Ca$^{2+}$ (*black*) and after sodium substitution (*green*). *p < 0.05; **p < 0.01****p < 0.0001.

DOI: https://doi.org/10.7554/eLife.40984.002

The following source data is available for figure 1:

**Source data 1.** Data plotted in *Figure 1*.

DOI: https://doi.org/10.7554/eLife.40984.003

In hippocampal neurons, a striking potentiation of the resting sodium leak conductance has been shown to occur under conditions of low extracellular calcium (*Chu et al., 2003*). The molecular basis of the potentiation involves relief of tonic inhibition from the calcium sensing receptor (CaSR) onto NALCN channels (*Lu et al., 2010*). In dopamine neurons, low external calcium results in faster spontaneous firing (*Khaliq and Bean, 2010*), but whether the increase in spontaneous firing is caused by potentiation of resting sodium leak current in low extracellular calcium solutions has not been tested.

We tested the effect of lowering extracellular calcium from 2 mM to 0.1 mM on the isolated sodium leak current. Switching to low calcium solution, the resting sodium leak current increased dramatically over a time-course of 6–10 min from an amplitude of $-50.69 \pm 5.35$ pA to $-192 \pm 20.07$ pA (n = 22, *Figure 1D–F*; p < 0.0001, RM one-way ANOVA followed by Tukey test). Blocking the resting sodium leak conductance with NMDG substitution reduced the holding current from $-192 \pm 20.07$ pA to $-64.53 \pm 8.96$ pA (n = 22; *Figure 1D–F*; p < 0.0001, RM one-way ANOVA followed by Tukey test). These recordings were made using $Cs^+$-based internal solutions, but in separate experiments we tested the amplitude of NMDG-sensitive currents in low extracellular calcium measured using $K^+$-based internal solutions. We observed no difference in the amplitude of NMDG-sensitive currents when recording with either $Cs^+$-based or $K^+$-based internal solutions (NMDG-sensitive current at $-70$ mV; $Cs^+$-based, $138 \pm 18$ pA, n = 22; $K^+$-based, $107.9 \pm 17$ pA, n = 8; p = 0.36).

To test directly for the involvement of NALCN channels in the low calcium potentiation of the resting leak, we recorded from dopaminergic neurons in NALCN conditional knockout mice. In knockout cells, there was relatively little effect of lowering extracellular calcium. Instead, we observed only a slight increase in $Na^+$ leak current from $-20.83 \pm 2.279$ to $-34.76 \pm 3.659$ pA (n = 9; p = 0.03, RM one-way ANOVA followed by Tukey test). Moreover, subsequent substitution of sodium with NMDG solution did not significantly reduce the holding current (normal $Na^+$, $-34.76 \pm 3.659$ pA; NMDG, $-24.58 \pm 4.72$ pA; n = 9, p = 0.14, RM one-way ANOVA followed by Tukey test) (*Figure 1G–I*). Therefore, in agreement with the work of Ren and colleagues in hippocampal neurons (*Lu et al., 2010*), our experiments in dopaminergic neurons demonstrate that the low calcium potentiation of the sodium leak reflects potentiation of current flowing through NALCN channels.

## Dopamine D2 receptors robustly inhibit NALCN current

D2 receptors play a dominant role in shaping the activity of midbrain dopamine neurons (*Anzalone et al., 2012*; *Bello et al., 2011*; *Gantz et al., 2013*). NALCN channels are activated by $G_q$ protein-coupled receptors (*Lu et al., 2009*), but it is unknown whether NALCN-mediated currents are modulated by $G_{i/o}$ coupled receptors. Therefore, we tested the effect of D2 receptor activation on the sodium leak current by bath applied dopamine (100 µM). GIRK channels are not $Cs^+$ permeable, suggesting that our $Cs^+$-based internal solution is sufficient to block most of GIRK currents (*Watts et al., 1996*). To ensure that GIRK channels were blocked in this set of experiments, we added 100 µM $BaCl_2$ to the bath solution.

Activation of D2 receptors with dopamine resulted in a dramatic reduction in the amplitude of the sodium leak current. Bath application of dopamine led to inhibition of the leak current from $-201.1 \pm 19.03$ pA under control conditions to $-133.9 \pm 14.68$ pA at the time of maximal dopamine inhibition (n = 14; p < 0.0001, RM one-way ANOVA followed by Tukey test) (*Figure 2A,G and H*). To determine the fraction of the sodium leak conductance that was inhibited by dopamine, we then substituted sodium for NMDG to block sodium leak current. Subsequent NMDG substitution did not further reduce holding current (dopamine, $-133.9 \pm 14.68$ pA; NMDG, $-116.2 \pm 13.31$ pA, n = 14; p = 0.43, RM one-way ANOVA followed by Tukey test) (*Figure 2A and G*). We found that $88.08 \pm 8.36\%$ of the total sodium leak was inhibited by dopamine.

To test the involvement of NALCN in the D2 modulation of sodium leak, we repeated these experiments in NALCN knockout mice. The dopamine inhibition of the sodium leak was completely abolished in the NALCN conditional knockout mouse (low $Ca^{2+}$, $-57.94 \pm 11.62$ pA; dopamine, $-65.31 \pm 19$ pA; n = 7; p = 0.185, RM one-way ANOVA followed by Tukey test) (*Figure 2B,G and H*). Therefore, these experiments demonstrate that dopamine inhibits sodium leak current flowing through NALCN channels.

Next, we tested the concentration dependence of NALCN inhibition by dopamine. To generate a concentration-response curve, we tested the amplitude of sodium leak current inhibited by

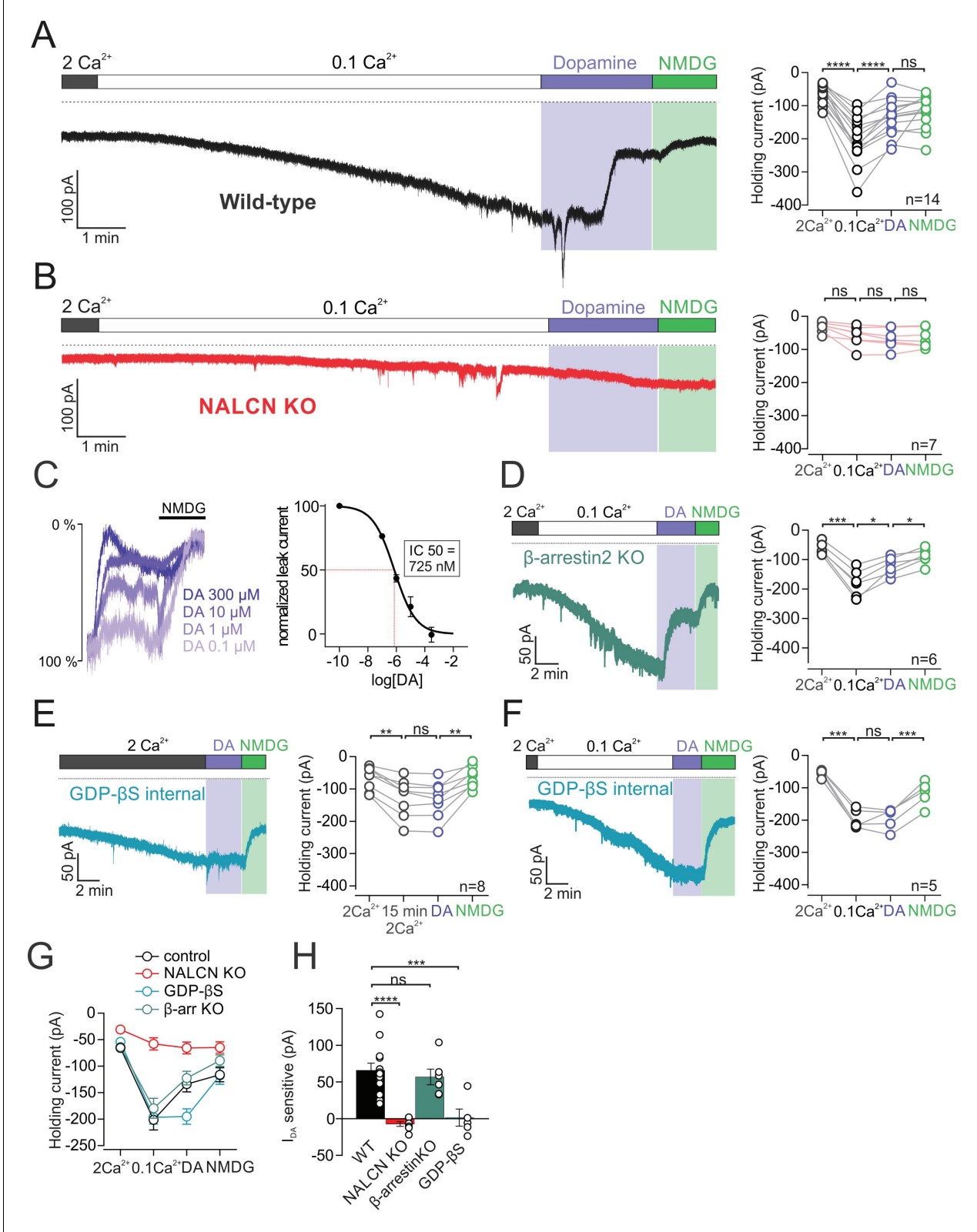

**Figure 2.** D2 receptor activation inhibits NALCN through a G-protein dependent mechanism. (**A**)*Left*, Example trace of isolated sodium leak current in wildtype dopamine neuron recorded at −70 mV in 2 mM $Ca^{2+}$ and 0.1 mM $Ca^{2+}$, followed by bath application of 100 µM dopamine and subsequent substitution of $Na^+$ with NMDG. *Right*, Summary plot of data in different conditions. (**B**) Example trace of isolated sodium leak current in NALCN cKO mice. Note, dopamine inhibition of sodium leak is abolished. (**C**) *Left*, Example traces showing inhibition of sodium leak current induced by 100 nM
*Figure 2 continued on next page*

*Figure 2 continued*

(n = 5), 1 µM (n = 3), 10 µM (n = 3), and 300 µM (n = 4) dopamine followed by block of leak current in NMDG solution. Traces have been normalized to control current amplitude. *Right*, Concentration-response curve for inhibition of leak current by dopamine. (**D**) Example trace of leak current in β-*arrestin2* KO mice. *Right*, Summary plot. (**E**) *Left*, Example trace of the isolated sodium leak current recorded with GDP-βS added to internal solutions. Dopamine applied to cell in 2 mM $Ca^{2+}$ external solutions. Note the slow increase in leak current with GDP-βS internal. *Right*, Summary plot. (**F**) Example trace of sodium leak current with GDP-βS internal solution. Dopamine applied to cell in 0.1 mM $Ca^{2+}$ external solutions. *Right*, Summary plot of data. (**G**) Summary plot of data shown in *A, B, D* and *E*. (**H**) Bar plot of the dopamine-sensitive current shown in *A, B, D* and *E*. *p < 0.05; **p < 0.01; ***p < 0.001; ****p < 0.001.

DOI: https://doi.org/10.7554/eLife.40984.004

The following source data is available for figure 2:

**Source data 1.** Data plotted in *Figure 2*.

DOI: https://doi.org/10.7554/eLife.40984.005

dopamine applied over a range of concentrations. Following application of dopamine in each cell, external sodium was then substituted with NMDG, which resulted in a maximal inhibition of sodium leak current. To calculate values for percent inhibition, the leak current inhibited by dopamine was normalized to the total sodium leak current amplitude, which is the leak current measured under controlled conditions minus the current recorded in NMDG. Examination of the concentration-response curve in *Figure 2C* shows that the concentration of half-maximal inhibition (IC50) of NALCN by dopamine was 725 nM (range, 491–961 nM; Hill coefficient, −0.6; range, 0.4–0.7). This value is comparable to the published EC50 values of 155 nM and 233 nM for dopamine activation of GIRK channels (*Kim et al., 1995*; *Uchida et al., 2000*).

It is well established that G protein-coupled receptors, including D2 receptors, signal through G protein-dependent and G protein-independent mechanisms. We first sought to test if inhibition of the sodium leak current relied on β-*arrestin*, which functions independently of G proteins (*Beaulieu and Gainetdinov, 2011*). To examine whether β-*arrestin* is involved in the inhibition of NALCN, we recorded D2 receptor mediated signal in cells from *Arr2* knockout mice (*Figure 2D,G and H*). We found that dopamine inhibition of NALCN was still present in *Arr2* knockout mice with a reduction of the sodium leak current from −179.3 ± 18.72 to −122.5 ± 12.82 pA (n = 6; p = 0.012, RM one-way ANOVA followed by Tukey test). Comparing the amplitude of the dopamine-sensitive sodium leak current, we observed no difference between wild-type (67.24 ± 8.96 pA; n = 14) and *Arr2* knockouts (56.86 ± 10.75 pA; n = 6) (p = 0.866, one-way ANOVA followed by Tukey test).

We next investigated the G protein-dependence of the dopamine inhibition of NALCN (*Figure 2E–F*). To test this, we substituted the GTP contained in our intracellular solutions for 1 mM GDP-βS, a nonhydrolyzable form of GDP which prevents G protein signaling. The first set of experiments were performed with 2 mM $Ca^{2+}$ in the extracellular solution (*Figure 2E*). Upon breakthrough, cells that were recorded using GDP-βS internal solution showed a slow but steady increase in the inward leak current (1–2 min after breakthrough, −67.69 ± 12.75 pA;~15 min following breakthrough, −125.9 ± 20.73 pA; n = 8; p = 0.0027, RM one-way ANOVA followed by Tukey test) (*Figure 2E*). The enhancement in leak current is likely due to GDP-βS dependent disruption in signaling from calcium sensing (CaSR) receptors, which produce an incomplete but constitutive inhibition of NALCN-mediated leak currents in the presence of extracellular calcium (*Lu et al., 2010*).

Substitution of GTP with GDP-βS in the intracellular solution completely abolished the D2 receptor-mediated inhibition of the NALCN current. Internal GDP-βS blocked the effect of dopamine when recordings were made in external solution containing 2 mM $Ca^{2+}$ (control, −125.9 ± 20.73 pA; dopamine, −131.2 ± 19.9 pA; n = 8; p = 0.4196, RM one-way ANOVA followed by Tukey test) (*Figure 2E*). Similar results were obtained in recordings made in 0.1 mM external $Ca^{2+}$ solution (control, −196.4 ± 12.07 pA; dopamine, −195.0 ± 14.72 pA; n = 5; p = 0.9992, RM one-way ANOVA followed by Tukey test) (*Figure 2F*). All together, these results demonstrate that D2 receptors inhibit NALCN in a G protein-dependent manner.

## D2 receptor modulation of NALCN channels slows firing independent of signaling through GIRK channels

The small amplitude of the sodium leak currents recorded in 2 mM external calcium raises the question of whether inhibition of NALCN by dopamine would have a significant physiological effect on

firing in the absence of GIRK activation (holding current at −70 mV; wild-type, −55.74 ± 4.29 pA, n = 39; NALCN KO, −19.69 ± 3.72 pA, n = 20; p < 0.0001, Mann-Whitney U-test) (*Figure 3A*). To test this, we first compared spontaneous activity recorded in cell-attached configuration in neurons from wild-type and NALCN knockout mice (*Figure 3B*). In wild-type animals, we found that 135 out of 148 of dopaminergic neurons (91.21%, 33 mice) were spontaneously active with firing frequencies above 0.1 Hz. In NALCN conditional knockout mice, however, we were surprised to find that only a minor fraction of dopamine neurons, 18 out of 60 (30%, 18 mice) were spontaneously active (*Figure 3C*). We confirmed that silent neurons were dopaminergic cells by breaking through and checking for the presence of a sag potential (I$_h$). In these silent dopaminergic neurons, the resting membrane potential immediately following breakthrough was hyperpolarized at an average of −53.65 ± 0.71 mV (n = 15). However, small current injections of 10–20 pA in these cells was sufficient to restore a regular tonic firing activity at 0.5 to 5 Hz (*Figure 3B*), indicating that these cells were healthy and capable of firing action potentials. These data show that although NALCN currents are small in amplitude when recorded in physiological external calcium, they play a central role in generating the depolarization necessary for driving pacemaking in dopaminergic neurons.

We next examined the impact of dopamine inhibition of the NALCN current on spontaneous firing. To minimize non-specific effects due to leaky pipette seals, we first recorded in cell-attached configuration. Under control conditions, we found that puff application of dopamine (300 µM, 1 s) resulted in a long duration pause in firing (47.88 ± 10.28 s, n = 8) as expected. To isolate effects mainly due to inhibition of NALCN channels alone, we then applied dopamine in the presence of extracellular BaCl$_2$ (0.1 mM) plus the high affinity peptide blocker tertiapin-Q (300 nM), which are both effective blockers of GIRK channels (*Jin and Lu, 1998*; *Sodickson and Bean, 1996*). In the presence of GIRK channel blockers, dopamine application resulted in a substantial reduction in firing from 2.56 ± 0.18 to 0.89 ± 0.26 Hz (n = 8; p = 0.0078, Wilcoxon test) (*Figure 3D,E*). We obtained similar results using a high concentration (0.5 mM) of extracellular BaCl$_2$ alone which was shown to completely block GIRK channels (*Sodickson and Bean, 1996*). Under these conditions, puff application of dopamine resulted in a 73.81 ± 10.47% reduction in the firing (n = 5) (*Figure 3—figure supplement 1*). Lastly, dopamine puff in solutions containing 300 nM tertiapin-Q only without added barium resulted in similar inhibition as well (n = 5) (*Figure 3—figure supplement 1*), suggesting that low concentrations of barium present in the extracellular solution do not significantly contribute to the observed effects.

The lack of spontaneous activity in neurons from NALCN knockout mice precluded cell-attached experiments examining effects of dopamine on spontaneous firing. Therefore, we performed a parallel set of experiments in whole-cell configuration where we drove tonic firing with small current injections. Similar to our cell-attached recordings, dopamine reduced firing in wild-type mice even in the presence of GIRK channel blockers 3.71 ± 0.32 to 1.71 ± 0.32 Hz (n = 9; p = 0.0039, Wilcoxon test) (*Figure 3F,G*). By contrast, dopamine neurons recorded in the NALCN conditional knockout showed no reduction in firing when dopamine was applied in the presence of BaCl$_2$ and tertiapin-Q. In fact, we observed a trend toward a small increase in the firing rate following puff application of dopamine (NALCN cKO mice; control, 3.024 ± 0.22 Hz; dopamine, 3.31 ± 0.25 Hz; n = 8; p = 0.078, Wilcoxon test) (*Figure 3F and G*). This increase in firing rate is similar to the excitation seen following activation of dopamine transporters (*Ingram et al., 2002*). Therefore, these results demonstrate that NALCN channels contribute significantly to pacemaking in dopamine neurons. Moreover, we show that even under conditions where GIRK currents are blocked, spontaneous firing is slowed by dopamine inhibition of NALCN channels alone.

## NALCN-mediated sodium leak currents are inhibited by activation of GABA-B receptors

The experiments described above revealed NALCN as a major ion channel effector for D2 receptor regulation of dopamine neuron activity. To extend this idea, we hypothesized that inhibition of NALCN channels may be part of a more general mechanism for modulation involving other G$_{i/o}$ protein-coupled receptors. GABA-B receptors also trigger GIRK channel opening and are expressed in dopamine neurons (*Labouèbe et al., 2007*; *Lüscher and Slesinger, 2010*). Therefore, we tested whether their activation would produce a similar inhibition of the NALCN-mediated sodium leak current.

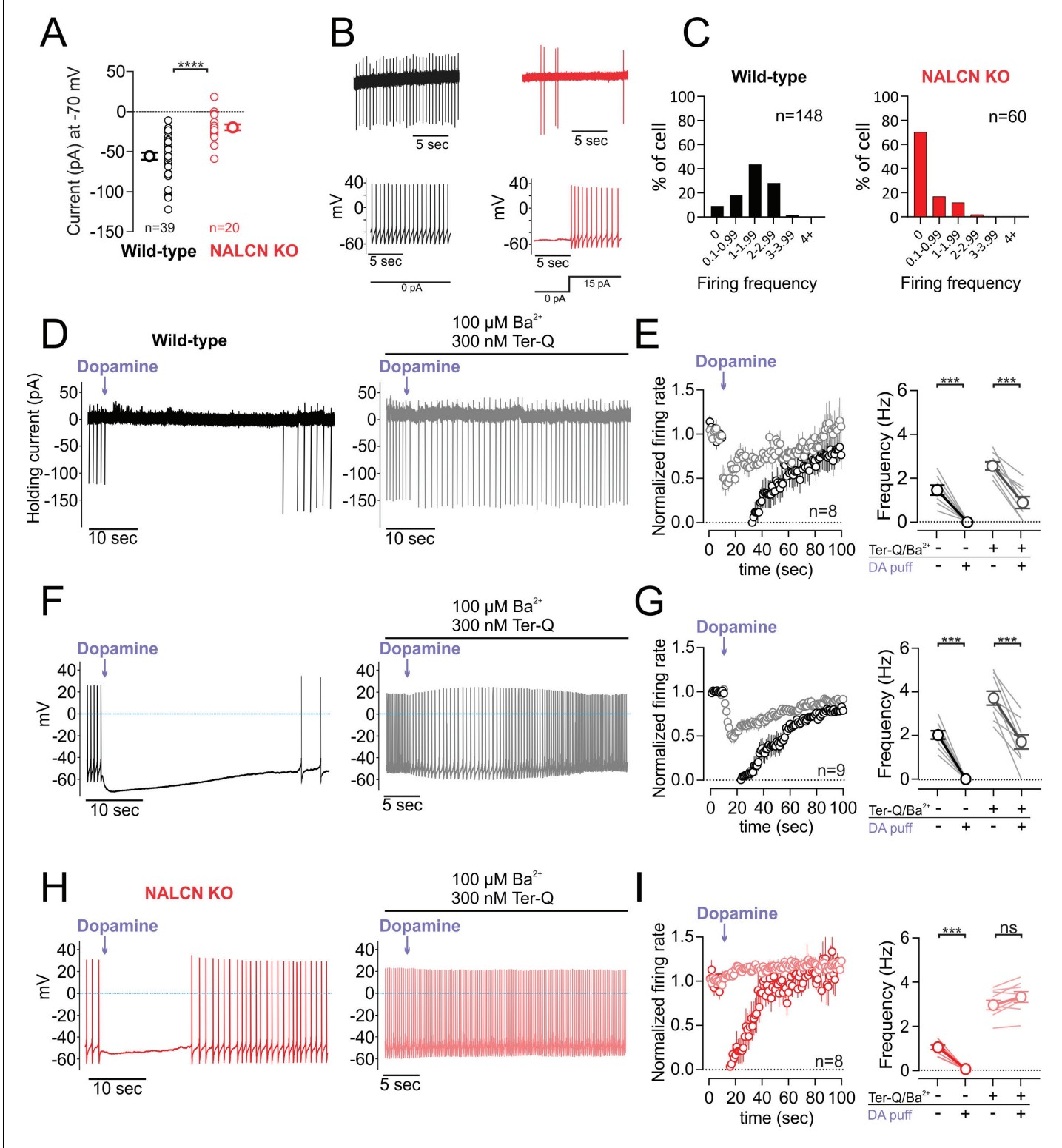

**Figure 3.** D2 receptor inhibition of NALCN slows spontaneous firing. (**A**) Summary plot of isolated sodium leak currents measured at −70 mV in wildtype (*black*) versus NALCN cKO mice (*red*). (**B**) Cell-attached (*top*) and whole-cell (*bottom*) recordings of pacemaking from dopamine neurons recorded in wildtype (*black*) and NALCN cKO mice (*red*). (**C**) Histogram plots of firing rates obtained during cell-attached recordings in wildtype and NALCN cKO mice. (**D**) Effect of dopamine puff (300 μM, 1 s) on the firing rate of dopamine neuron recorded in cell-attached mode with 2 mM external Ca$^{2+}$, before (*black*) and after block of GIRK channels by bath application of Ba$^{2+}$ and tertiapin-Q (*gray*). (**E**) *Left*, Averaged normalized firing frequency plotted against time showing the effect of dopamine puff in control conditions (*black*) and after the block of GIRK channels (*gray*). *Right*, Averaged

*Figure 3 continued on next page*

*Figure 3 continued*

absolute firing rate before and after dopamine puff in control conditions and after the block of GIRK channels. (**F,G**) Same experiment as *D* and *E* but in whole-cell configuration. (**H**) and (**I**) Same as *F* and *G* in NALCN cKO mice. ***p < 0.001.

DOI: https://doi.org/10.7554/eLife.40984.006

The following source data and figure supplements are available for figure 3:

**Source data 1.** Data plotted in *Figure 3*.
DOI: https://doi.org/10.7554/eLife.40984.011
**Figure supplement 1.** Effect of D2 receptor-mediated inhibition of NALCN on spontaneous firing recorded in the presence of different GIRK blocker combinations.
DOI: https://doi.org/10.7554/eLife.40984.007
**Figure supplement 1—source data 1.** Data plotted in *Figure 3—figure supplement 1*.
DOI: https://doi.org/10.7554/eLife.40984.008
**Figure supplement 2.** Lack of correlation between dopamine inhibition of firing and the rate of spontaneous firing.
DOI: https://doi.org/10.7554/eLife.40984.009
**Figure supplement 2—source data 1.** Data plotted in *Figure 3—figure supplement 2*.
DOI: https://doi.org/10.7554/eLife.40984.010

Activation of GABA-B receptors with baclofen resulted in a substantial inhibition of sodium leak current. Recording the isolated sodium leak current at $-70$ mV in low calcium, we found that bath application of baclofen 10 µM reduced the holding current from $-185 \pm 16.06$ pA to $-106 \pm 13.89$ pA in baclofen (n = 8; p = 0.0041, RM one-way ANOVA followed by Tukey test) (*Figure 4A*). Furthermore, the baclofen-sensitive current was similar in amplitude to the dopamine-sensitive current (baclofen-sensitive current, $79 \pm 14.13$ pA, n = 8; dopamine-sensitive current, $67.24 \pm 8.96$ pA, n = 14; p = 0.66, Mann-Whitney U-test). Immediately following baclofen application, we washed in a low sodium NMDG external solution to block the sodium leak current that remained in baclofen. We found that following baclofen application, substitution of sodium with NMDG produced no further block of the sodium leak currents (p = 0.21, RM one-way ANOVA followed by Tukey test) indicating that activation of GABA-B receptors results in nearly complete inhibition of sodium leak currents. Similar to our results examining dopamine, the baclofen inhibition of the isolated sodium leak current recorded at $-70$ mV was completely abolished in cells from NALCN conditional knockout mice (low $Ca^{2+}$, $-54.47 \pm 13.57$ pA; baclofen, $-61.89 \pm 12.52$ pA, n = 7, p = 0.0436, RM one-way ANOVA followed by Tukey test) (*Figure 4B*), demonstrating that GABA-B receptors inhibit sodium leak currents flowing through NALCN channels.

## GABA-B receptor modulation of NALCN channels slows firing independent of GIRK channel signaling

To better understand the functional impact of GABA-B modulation through NALCN, we tested the effect of the GABA-B receptor-mediated inhibition of NALCN channels on spontaneous firing in dopamine neurons. In control conditions, puff application of baclofen (10 µM, 1 s) paused the firing as expected. In the presence of $BaCl_2$ and tertiapin-Q, a significant decrease of the firing was induced by the baclofen puff in wild-type (from $3.67 \pm 0.29$ to $0.76 \pm 0.37$ Hz; n = 8, p = 0.0002, Wilcoxon) (*Figure 5A,B*). By contrast, $BaCl_2$ and tertiapin-Q completely abolished the effect of baclofen on firing in dopamine neurons recorded in NALCN conditional knockout mice (frequency, control, $3.02 \pm 0.71$ Hz; baclofen, $3.02 \pm 0.66$ Hz; n = 8; p > 0.99, Wilcoxon) (*Figure 5C,D*).

## Baclofen activation of GABA-B receptors potently inhibits NALCN channels

Past studies have tested the sensitivity of GIRK channels to baclofen and have reported EC50 values of 9.2 µM and 14.8 µM (*Chan et al., 1998*; *Cruz et al., 2004*). By contrast, our experiments above show that 10 µM baclofen produced a near complete inhibition of sodium leak current (*Figure 4A*), suggesting a much higher potency of baclofen in the modulation of NALCN channels. Therefore, we tested the sensitivity of the NALCN current to baclofen (*Figure 6A,B*). Interestingly, our data showed inhibition of NALCN current by baclofen concentrations as low as 100 nM. Concentration-dependence curves yielded an IC50 value for baclofen of 267 nM (range, 234–301 nM; Hill

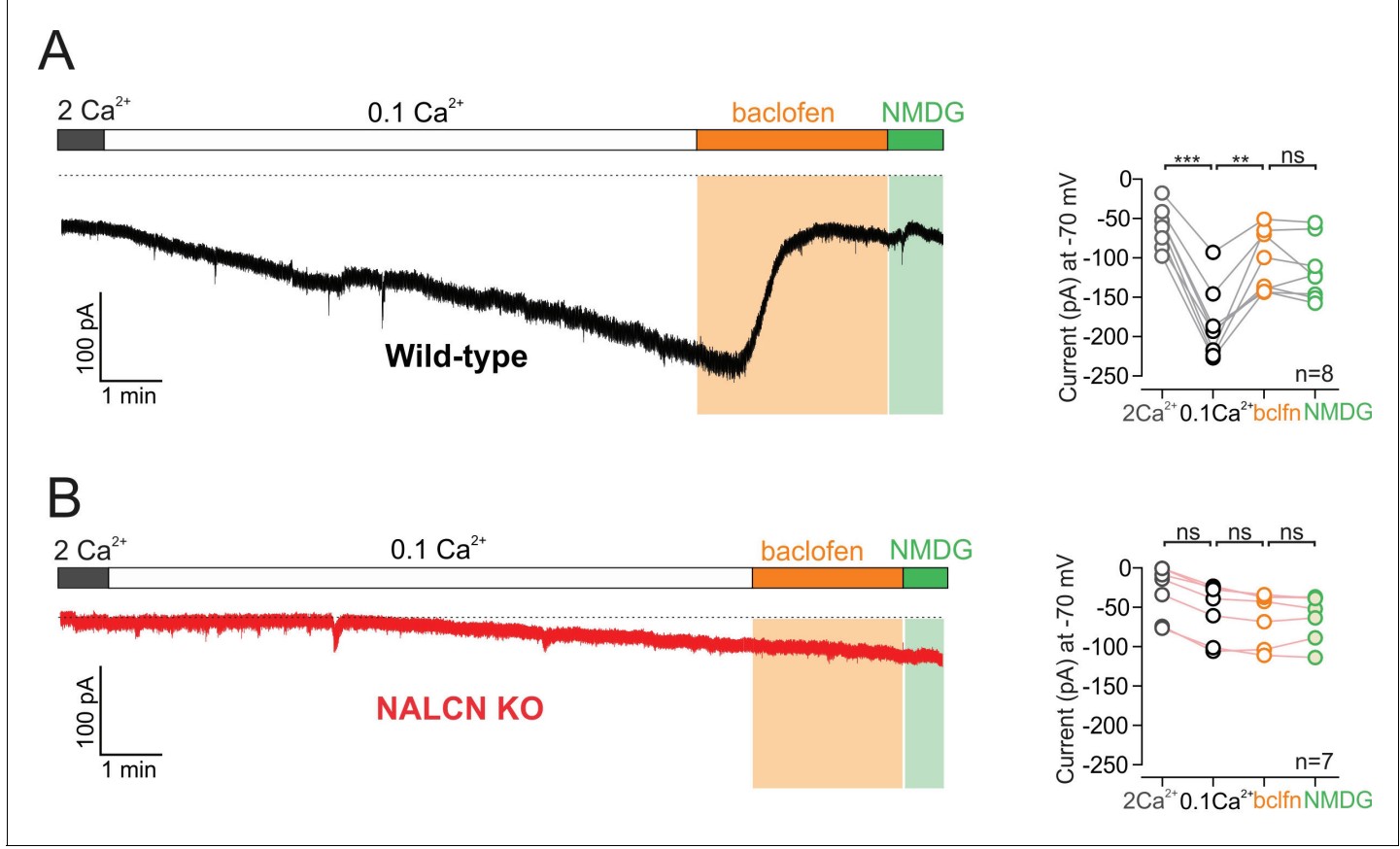

**Figure 4.** GABA-B receptor activation inhibits NALCN. (**A**) *Left*, Example trace of showing effect of 10 μM baclofen on isolated sodium leak current recorded in wildtype mice. Note, Na$^+$ substitution with NMDG produces no further effect following baclofen. *Right*, Summary plot. (**B**) Example trace of showing effect of baclofen on sodium leak current recorded in NALCN cKO mice.

DOI: https://doi.org/10.7554/eLife.40984.012

The following source data is available for figure 4:

**Source data 1.** Data plotted in *Figure 4*.

DOI: https://doi.org/10.7554/eLife.40984.013

coefficient, −1.1; range, 0.9–1.3), more than an order of magnitude lower than the published EC50 value for baclofen activation of GIRK.

Given the high sensitivity of NALCN to baclofen, we asked whether firing in the dopamine neurons would be inhibited by baclofen concentrations near or below the minimum necessary to activate substantial GIRK current (*Figure 6C–E*). Therefore, we tested bath application of 300 nM baclofen on a background of tertiapin-Q to block GIRK. Baclofen delivered at 300 nM reduced the firing frequency of dopamine neurons by 44.79%, from 2.39 ± 0.41 Hz to 1.07 ± 0.28 Hz (n = 7, p = 0.0039, one-way ANOVA followed by Tukey test). Subsequent application of the GABA-B specific blocker, CGP 55845 (1 μM), restored cells to their initial firing rate. Therefore, these results demonstrate that baclofen delivered at concentrations that are below the range necessary to activate GIRK channels can significantly inhibit excitability, primarily through inhibition of NALCN.

## Discussion

These data demonstrate that NALCN is a major ionic contributor to the generation of spontaneous activity in midbrain dopaminergic neurons. Furthermore, we provide the first evidence that the NALCN-mediated sodium leak conductance is negatively modulated by both dopamine D2 receptors and GABA-B receptors. We also show that modulation of NALCN leads to significant slowing of spontaneous activity, consistent with its role in driving pacemaking. Lastly, we show that baclofen

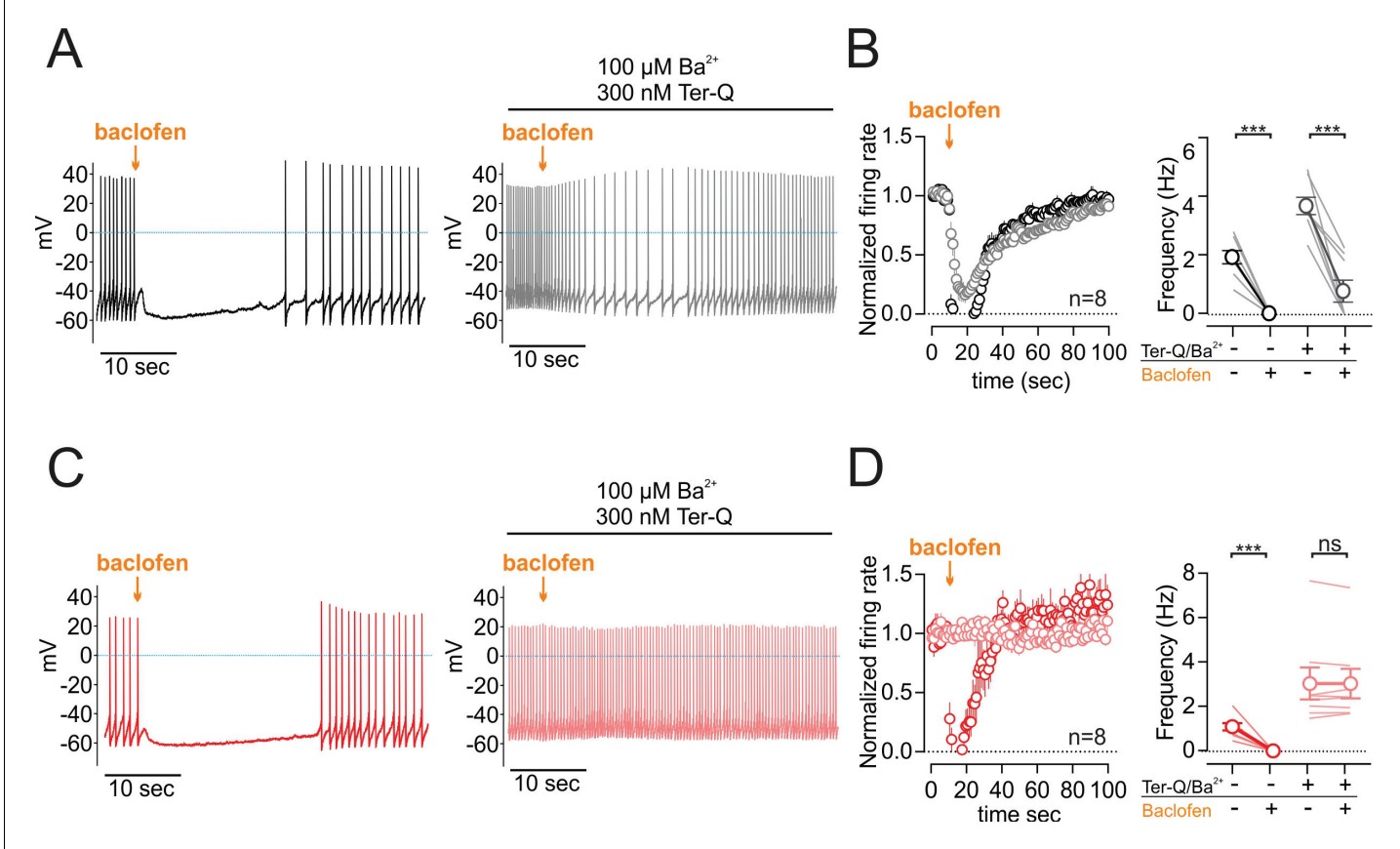

**Figure 5.** GABA-B receptor inhibition of NALCN slows spontaneous firing. (**A**) Effect of baclofen puff (10 µM, 1 s) on the firing rate of dopamine neuron recorded in whole-cell configuration with 2 mM external $Ca^{2+}$, before (*black*) and after block of GIRK channels by $Ba^{2+}$ and tertiapin-Q (*gray*). (**B**) *Left*, Averaged normalized firing frequency plotted against time showing the effect of 10 µM baclofen puff in control (*black*) and after the block of GIRK channels (*gray*). *Right*, Absolute firing rates before and after baclofen puff in control conditions and after the block of GIRK channels. (**C**) and (**D**). Same as *C* and *D* in NALCN cKO mice. **p < 0.01; ***p < 0.001.
DOI: https://doi.org/10.7554/eLife.40984.014

The following source data is available for figure 5:

**Source data 1.** Data plotted in *Figure 5*.
DOI: https://doi.org/10.7554/eLife.40984.015

exhibits a higher potency in inhibiting NALCN than for activating GIRK channels. Thus, we identify the NALCN channel as a core effector of $G_{i/o}$ protein-coupled receptors that functions in concert with GIRK channels to inhibit neuronal firing in midbrain dopaminergic neurons.

## Contribution of NALCN to pacemaking in dopaminergic neurons

It has been long appreciated that sodium leak currents are present in a variety of spontaneously active neurons. Voltage-clamp experiments using ion substitution have identified prominent sub-threshold sodium leak currents in dopamine neurons (*Khaliq and Bean, 2010*), GABAergic neurons of the substantia nigra pars reticulata (SNr) (*Atherton and Bevan, 2005*), neurons of the cerebellar nucleus (*Raman et al., 2000*) and suprachiasmatic nucleus neurons (*Jackson et al., 2004*). However, the lack of specific blockers coupled with imprecise knowledge of the channels that generate leak currents have complicated efforts to study their role in pacemaking.

Using NALCN conditional knockout mice, we provide clear evidence that pacemaking in dopaminergic neurons is driven in part by sodium leak current flowing through NALCN channels. Specifically, we found that knockout of NALCN in dopaminergic neurons results in significantly smaller resting sodium leak currents measured at −70 mV. Second, direct inhibition of NALCN by D2 and

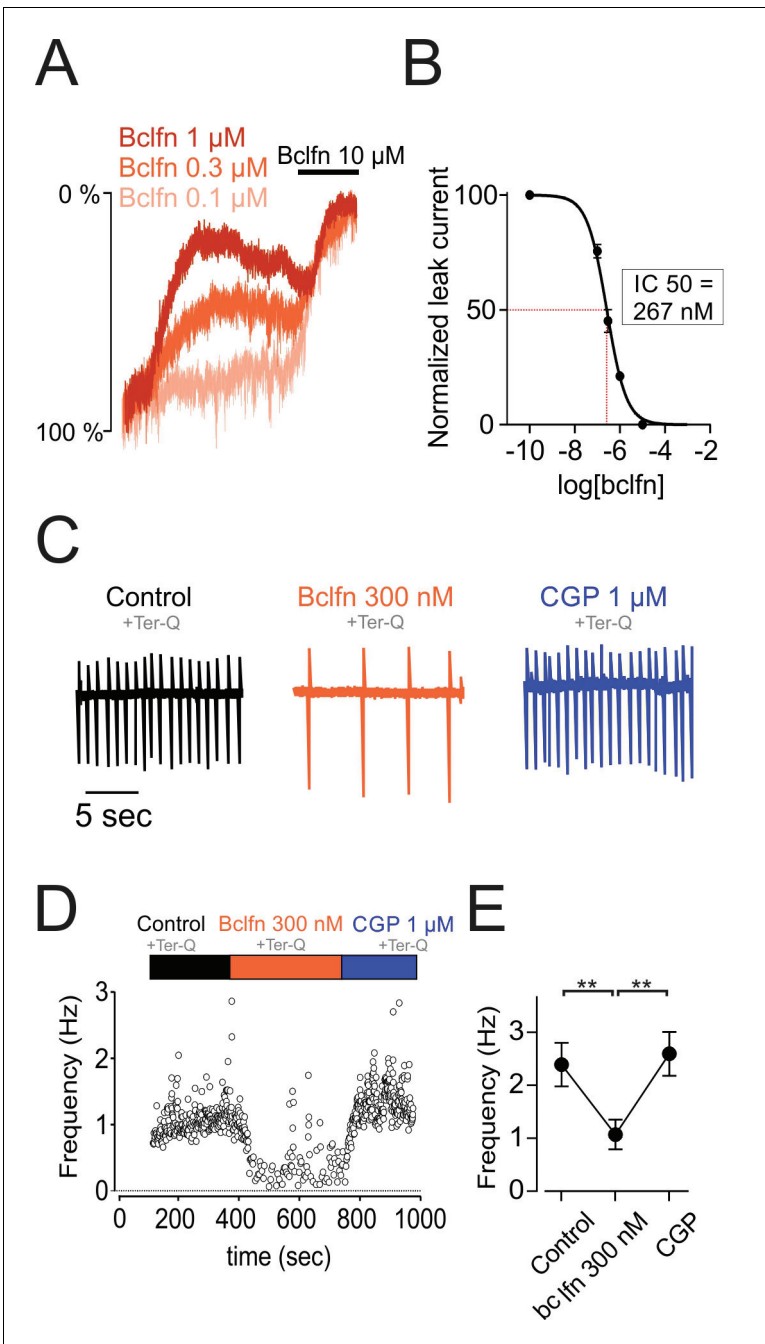

**Figure 6.** NALCN currents are highly sensitive to inhibition by GABA-B receptors. (**A**) Example traces showing inhibition of sodium leak current by 100 nM (n = 4), 300 nM (n = 5), 1 μM (n = 5), and 10 μM (n = 5) baclofen. Traces have been normalized to control current amplitude. (**B**) Concentration-response curve of NALCN channel block by baclofen. (**C**) Cell-attached recording in control conditions (*black*), in the presence of 300 nM baclofen (*orange*) and after blocking GABA-B receptors with 1 μM CGP 55845 (*blue*). All recordings were made in the presence of tertiapin-Q (300 nM). (**D**) Time course of the instantaneous firing rate for the cell shown in *C*. (**E**) Summary plot of firing frequencies of 7 cells in experiments similar to those shown in *C,D*. Averaged values are shown as closed symbols. **p < 0.01.

DOI: https://doi.org/10.7554/eLife.40984.016

The following source data is available for figure 6:

**Source data 1.** Data plotted in *Figure 6*.
DOI: https://doi.org/10.7554/eLife.40984.017

GABA-B receptors slows pacemaking. Lastly, the majority of SNc neurons (70%) recorded in brain slices from NALCN conditional knockout mice lack spontaneous activity. Consistent with these findings, recent studies in SNr GABAergic neurons and chemosensory neurons of the retrotrapezoid nucleus have found that knockout of NALCN results in hyperpolarization of the resting membrane potential and a ~ 50% reduction in spontaneous firing rate (*Lutas et al., 2016*; *Yeh et al., 2017*). In circadian pacemaker neurons, knockout of NALCN leads to nearly complete abolishment of pacemaking (*Flourakis et al., 2015*). In agreement with these observations, therefore, our results in dopaminergic neurons suggest that NALCN plays a central role in pacemaking.

A previous study of the conductances that drive pacemaking reported that background sodium leak currents in VTA dopamine neurons are large in amplitude, while sodium leak currents in SNc neurons are relatively small (*Khaliq and Bean, 2010*). Given this prior result, we were surprised to see that the majority of SNc neurons in NALCN knockout mice lack spontaneous firing, and that D2R and GABA-BR modulation of the NALCN current produces such a significant inhibition of pacemaking. Although we also observed small amplitude sodium leak currents with 2 mM external calcium (*Figure 1B*), this study focuses on G protein receptor modulation of spontaneous firing which is highly sensitive to small conductance changes. In addition to the previously described differences in leak currents between VTA and SNc neurons (*Khaliq and Bean, 2010*), however, heterogeneity may also exist across subpopulations of SNc neurons either in NALCN expression and/or differences in the regulation of NALCN by external calcium through the calcium sensing receptor (CaSR). Therefore, it will be important for future studies to examine the expression and regulation of NALCN in midbrain dopamine neurons subpopulations.

The absence of spontaneous activity in dopamine neurons from the NALCN knockout mice was also surprising given the known robustness of firing. Action potential firing is generate by multiple subthreshold conductances with overlapping functions, an arrangement that favors compensatory adaptation when single conductances are inhibited (*Drion et al., 2011*; *Guzman et al., 2010*; *Kimm et al., 2015*; *Marder and Goaillard, 2006*; *Swensen and Bean, 2005*). In addition, homeostatic mechanisms can compensate for genetic ablation of ion channels in knockout mice. As an example, $Ca_V1.3$ L-type calcium channels are active during pacemaking in dopamine neurons, but knockout of $Ca_V1.3$ has little effect on firing (*Blythe et al., 2009*) due to upregulation of T-type channels (*Poetschke et al., 2015*). In the NALCN knockout mice, by contrast, autonomously-generated spontaneous firing was absent in most dopaminergic neurons. It is possible that homeostatic compensation was not sufficient to restore firing, which may be evidence for the critical role of NALCN in pacemaking. Alternatively, compensation may have occurred at the level of synaptic drive onto the dopamine neurons. Therefore, future experiments should test whether compensatory adaptations have occurred in excitatory synaptic signaling.

## $G_{i/o}$ protein-coupled receptor modulation of NALCN

We provide the first direct evidence of $G_{i/o}$ protein-coupled receptor modulation of NALCN. However, G protein-coupled receptors are known to influence multiple effector targets which raises the question of whether other channels may contribute to the effects described here. For example, GABA-B receptors in hippocampus and entorhinal cortex have been shown to activate background leak tandem-pore potassium (K2P) channels, TREK1 and 2 (*Breton and Stuart, 2017*; *Deng et al., 2009*; *Sandoz et al., 2012*). In VTA dopamine neurons recorded in GIRK2 knockouts, activation of GABA-B receptors evokes a $Ba^{2+}$ insensitive, non-GIRK current that was hypothesized to be produced by K2P channels (*Cruz et al., 2004*). Importantly in SNc neurons, we found the that $Ba^{2+}$-insensitive effects of GABA-B and D2 receptors observed in wildtype mice were abolished in NALCN knockout mice. These observations suggest that differences may exist between GABA-B signaling in SNc and VTA dopamine neurons.

Our data show that the sensitivity of NALCN to baclofen is high, with an IC50 of 267 nM. By comparison, GIRK channels exhibit a much lower sensitivity to baclofen according to published EC50 values in dopamine neurons (9.2 and 14.8 µM) (*Chan et al., 1998*; *Cruz et al., 2004*) and in other neurons such as midbrain GABAergic neurons and hippocampal neurons (range, 0.9–4.5 µM) (*Chan et al., 1998*; *Cruz et al., 2004*; *Sodickson and Bean, 1996*). The functional importance of this difference is currently unknown. However, one possibility is that the proximity of GABA-B receptor to inhibitory synapses may determine the extent to which NALCN or GIRK channels are recruited. GABA-B receptors located close to synapses are exposed to high GABA concentrations which would

favor GIRK activation. On the other hand, modulation through extrasynaptic GABA-B receptors, which are activated by lower concentrations of GABA, may primarily involve inhibition of NALCN.

The $G_{i/o}$ protein-coupled receptor inhibition of NALCN may add new features to signal processing mechanisms in dopamine neurons. For instance, GIRK and NALCN could potentially play a synergistic role in membrane hyperpolarization. While both hyperpolarize the membrane potential, activation of GIRK channels decreases input resistance while inhibition of NALCN increases input resistance. As a result, simultaneous inhibition of NALCN may function to potentiate GIRK-mediated hyperpolarization. Moreover, the tightened membrane may allow synaptic inputs to drive rebound firing more effectively from hyperpolarized membrane potentials (*Evans et al., 2017*).

The spatial proximity of NALCN and GIRKs could also enable potential interactions between these channels. Indeed, we find that NALCN currents are inhibited nearly maximally by activation of either GABA-B and D2 receptors (see *Figures 2A* and *4A*). This suggests an interesting scenario where GABA-B and D2 receptors may be arranged spatially into overlapping microdomains in which both receptors have access to GIRK channels and the NALCN signaling complex. Likewise, it has been shown that GIRK channels are inhibited by intracellular calcium but enhanced by internal sodium ions (*Kramer and Williams, 2016*; *Wang et al., 2016*). This raises the question of whether NALCN and GIRK channels colocalize in a complex with $G_{i/o}$ protein-coupled receptors and whether $Na^+$ flowing through NALCN channels potentiates GIRK currents. Future studies should focus on determining the subcellular locations of NALCN, GIRKs and other effectors of $G_{i/o}$ protein-coupled receptors.

As mentioned, existing evidence indicates that the mechanisms of GABA-B modulation may differ substantially between subpopulations of VTA dopamine neurons and SNc neurons. Specifically, VTA dopamine neurons that project to medial and lateral nucleus accumbens differ markedly in their responses to baclofen (*Yang et al., 2018*). In addition, past work has shown that the expression level of GIRK2 mRNA is 10 times lower in dopaminergic neurons of the medial VTA relative to laterally located mesostriatal dopamine neurons (*Lammel et al., 2008*). Interestingly, despite the low expression of GIRK expression in mesocortical cells, it was shown that bath application of both GABA and baclofen silenced the firing of mesocortical dopamine neurons completely. Our data demonstrate that inhibition of NALCN channels alone is sufficient to slow down the firing rate. Therefore, it will be important to determine the extent to which inhibition of NALCN contributes to the GIRK-independent GABA-B inhibition of medial VTA dopamine neurons.

Collectively, our findings provide evidence that NALCN is an effector of $G_{i/o}$ coupled receptors in dopamine neurons. Both GABA-B and D2 receptors modulate NALCN, raising the possibility that this may be a common feature to other $G_{i/o}$ protein-coupled receptors across different neuronal cell types. These results may provide a novel avenue for drug development strategies and molecular therapeutics that target Gi/o protein-coupled receptor signaling pathways.

## Materials and methods

### Animal use and generation of transgenic mice

All procedures were carried out in accordance with guidelines set by the animal care and use committee for the National Institute of Neurological Disorders and stroke and the National Institutes of Health. Adult (2–5 months) tyrosine hydroxylase-GFP (Th-GFP; C57BL/6 background) (*Matsushita et al., 2002*), *Arr2* knockout (Jackson Labs) and NALCN conditional knockout mice (C57BL/6 background) of either sex were used for electrophysiology. We generated mice in which positions between exons 5 and 6 of the *NALCN* allele was flanked by lox-P sites (*Nalcn^flox^*). To do this, we acquired ES cells from KOMP (UC Davis). ES Cells were microinjected into a mouse blastocyst and implanted into a female mouse (NIMH/NINDS Transgenic Core). Chimeras were born and bred to a *Flp-deleter* strain (C57BL/6 background; Jackson Labs) to make conditional ready mice *Nalcn^flox^*. *Nalcn^flox^* mice were then bred to *Slc6a3-Cre* mice, resulting in offspring (*Nalcn^flox/flox^*; *Slc6a3^Cre^*) that lack expression of NALCN in dopaminergic neurons. A recent publication reported *Nalcn^flox^* mice that were generated independently using ES cells obtained from the same source (KOMP), with exon deletion sites that are exactly the same as those reported in this study (*Yeh et al., 2017*).

## Slice preparation

Mice were anesthetized with isoflurane and transcardially perfused with an ice-cold glycerol-based slicing solution containing (in mM): 198 glycerol, 2.5 KCl, 1.2 NaHPO$_4$, 10 HEPES, 21 NaHCO$_3$, five glucose, 2 MgCl$_2$, 2 CaCl$_2$, 5 Na-ascorbate, 3 Na-pyruvate and two thiourea. Coronal 250 μm thick slices were then cut using a DTK-ZERO1 Microslicer and incubated at 34°C for 35 min and then stored at room temperature in a holding solution containing (in mM): 92 NaCl, 30 NaHCO$_3$, 1.2 NaH$_2$PO$_4$, 2.5 KCl, 35 glucose, 20 HEPES, 2 MgCl$_2$, 2 CaCl$_2$, 5 Na-ascorbate, 3 Na-pyruvate, and two thiourea. Recordings 40 min to 6 hr after being removed from the bath.

## Electrophysiological recordings

For patch-clamp recordings, slices were placed into a recording chamber and continuously superfused with warm (34°C) recording solution of the following composition (in mM): 125 NaCl, 25 NaHCO$_3$, 1.25 NaH$_2$PO$_4$, 3.5 KCl, 10 glucose, 1 MgCl$_2$, and 2 CaCl$_2$ (unless otherwise indicated). Neurons were visualized using a BX51WI Olympus microscope equipped with a CCD camera (W105AE, Watec). Patch-clamp recordings were obtained using low-resistance pipettes (3–5 MΩ) that were pulled from filamented borosilicate glass with a flaming/brown micropipette puller (Sutter Instruments). In voltage-clamp experiments, the internal solution contained the following (in mM): 122 CsMeSO$_3$, 9 HEPES, 1.8 MgCl$_2$, 4 Mg-ATP, 0.3 Na-GTP, 14 phosphocreatine, 0.45 EGTA and 0.09 CaCl$_2$. Current-clamp recordings were performed using an internal solution containing (in mM): 122 KMeSO$_3$, 9 NaCl, 9 HEPES, 1.8 MgCl2, 4 Mg-ATP, 0.3 Na-GTP, 14 phosphocreatine, 0.45 EGTA and 0.09 CaCl2. Salts were purchased from Sigma-Aldrich (St-Louis, MO).

Cell-attached recordings for the agonist 'puff experiments' shown in *Figures 3* and *6* were performed in a loose-seal voltage-clamp configuration (holding at −60 mV) using our normal extracellular recording solutions. The puffer pipette was positioned near cells, just above the slice to prevent direct mechanical effects of puff application. A subset of the spontaneous firing rates reported in *Figure 3C* were obtained from cell-attached recordings made using high-resistance seals. We routinely recorded cell-attached firing prior to breakthrough. These recordings were performed using either Cs-based or K-based intracellular solutions, as described above.

Low sodium solution experiments were performed using extracellular solutions in which 125 mM NaCl was replaced by N-methyl-D-glucamine (NMDG)-Cl. Solutions were made by first adding (in mM) 125 NMDG, 25 NaHCO$_3$ and 1.25 NaH$_2$PO$_4$ to the beaker. Next, the NMDG solution was titrated with HCl to a pH of 7.3–7.4. We then added 10 glucose and then bubbled the solution with 95/5% O$_2$/CO$_2$ to saturation. Last, either 1 MgCl$_2$ alone or 1 MgCl$_2$ plus 2 CaCl$_2$ were added. The osmolarity of both high and low Na external solutions was typically in the range of 300–315 mOsM.

Signals were digitized with a Digidata 1440A interface, amplified using a Multiclamp 700B amplifier and acquired using pClamp 10 software (Molecular Devices, Sunnyvale, CA). Data were sampled at 20 kHz and filtered at 10 kHz. Recordings were post hoc filtered at 1 kHz. Reported voltages were not corrected for liquid junction potential, which in our MeSO$_3$ based solutions measured at −8 mV.

All recordings were performed on dopamine neurons from the substantia nigra. Dopamine neurons were first targeted by their location and their large cell bodies. They were then identified based on various electrophysiological characteristics such as the firing frequency (<5 Hz), the presence of Ih and the GFP fluorescence in TH-GFP mice. In gap-free voltage-clamp experiments, a 10 mV step was applied each time a different solution entered in the bath allowing us to evaluate the access resistance of the cell.

## Drugs

Voltage-clamp and current-clamp recordings were performed in the continuous presence of synaptic blockers (20 μM CNQX, 50 μM APV, 50 μM picrotoxin). For the voltage-clamp experiments, 1 μM tetrodotoxin, 100 μM BaCl$_2$, 300 nM apamin and 3 mM Cs$^+$ were added to the bath. For experiments examining the concentration-dependence of dopamine (*Figure 3C*) were recorded with 1 μM CGP 55845, 1 μM SCH 39166, and 50 μM nomifensin were added to the bath. Experiments examining concentration dependence baclofen as well as accompanying cell-attached recordings (all data shown in *Figure 6*) were made with 1 μM SCH 39166, and 50 μM nomifensin were added to the

bath. Drugs were purchased from Tocris Bioscience (Bristol,UK), except for tertiapin-Q which was purchased from Alomone (Jerusalem, Israel).

## Data analysis

Data were analyzed using both Prism (GraphPad software), Clampfit (Molecular Devices) and IGOR (Wavemetrics) and were expressed in mean ±SEM. Statistical significance was determined in two group comparisons by two-tailed Mann-Whitney U-test or Wilcoxon signed-rank test (paired comparisons) and in more than two group comparisons by one-way ANOVAs or one-way repeated measures ANOVAs (paired comparisons) followed by the Tukey's post hoc test. The difference was considered significant at $p < 0.05$. At least three animals were tested per condition.

In our analysis of the dopamine inhibition of NALCN (*Figure 2A*), we minimized effects of D2 receptor desensitization (*Gantz et al., 2015*) by comparing the amplitude of the leak current in control condition to the amplitude at maximal inhibition. For plots of the normalize firing rate (*Figure 3E,G,I*), firing rates from individual experiments were normalized to the baseline firing rate, which was obtained from 10 averages a period of 10 s before the puff of dopamine (300 µM, 1 s) or baclofen (10 µM, 1 s). The instantaneous firing rate was obtained by averaging the number of spikes that occur within 1 s window. Summary plots show baseline values, which are 10 s averages before the puff application, and the minimum firing rates following puff application of dopamine or baclofen.

## Acknowledgements

Funding for this research was supported by the National Institute of Neurological Disorders and Stroke Intramural Research Program Grant NS003135 to ZMK. We thank Jim Pickel at the NIMH Transgenic Core Facility for help in generating the NALCN conditional knockout mice. We thank Robert Scott and Jia-Hua Hu for their help with transgenic mice. We thank Sherry Zhang for technical assistance. We also thank Rebekah Evans, Paul Kramer and Emily Twedell for helpful comments on the manuscript.

## Additional information

### Funding

| Funder | Grant reference number | Author |
| --- | --- | --- |
| National Institute of Neurological Disorders and Stroke | NS003135 | Zayd M Khaliq |

The funders had no role in study design, data collection and interpretation, or the decision to submit the work for publication.

### Author contributions

Fabian Philippart, Conceptualization, Data curation, Formal analysis, Investigation, Methodology, Writing—original draft, Writing—review and editing, conducted the experiments and analyzed the data, designed the experiments and wrote the paper; Zayd M Khaliq, Conceptualization, Supervision, Funding acquisition, Investigation, Methodology, Writing—original draft, Project administration, Writing—review and editing, designed the experiments and wrote the paper

### Author ORCIDs

Zayd M Khaliq http://orcid.org/0000-0002-1445-1457

### Ethics

Animal experimentation: All procedures were carried out in accordance with guidelines set by the animal care and use committee for the National Institute of Neurological Disorders and stroke and the National Institutes of Health (ACUC protocol number 1332).

Decision letter and Author response
Decision letter https://doi.org/10.7554/eLife.40984.020
Author response https://doi.org/10.7554/eLife.40984.021

## Additional files

### Supplementary files

• Transparent reporting form
DOI: https://doi.org/10.7554/eLife.40984.018

### Data availability

Data used to generate summary plots presented in Figures 1-6 are included in the manuscript and are provided as source data files.

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
