## [Decision Letter]

Thank you for submitting your article "G_i/o_ protein-coupled receptors in dopamine neurons inhibit the sodium leak channel NALCN" for consideration by *eLife*. Your article has been reviewed by three peer reviewers, and the evaluation has been overseen by Gary Westbrook as the Senior and Reviewing Editor. The following individuals involved in review of your submission have agreed to reveal their identity: Michael Beckstead (Reviewer #1); Andrew Lutas (Reviewer #2). The reviewers have discussed the reviews with one another and the Senior Editor has drafted this decision to help you prepare a revised submission.

Summary:

The reviewers were intrigued by the manuscript, but had a number of mostly non-overlapping concerns that would in some cases require additional data to address. The reviewers each made extensive comments, but in our post-review discussion we agreed that we thought you would be able to address these issues with additional comments and some additional experiments within a timeframe of 2 months. Because of the explanatory text in each review, I thought it was more helpful to you to provide the text of the original reviews rather than attempt to consolidate the comments into a single set of major and minor comments.

*Reviewer #1:*

This interesting manuscript by Philippart and Khaliq identifies the sodium leak channel NALCN as a mediator of G_i/o_ receptor inhibition in midbrain dopaminergic neurons. In spite of the large body of literature describing D2 and GABAB receptor signaling in these neurons, the investigators (aided by a NALCN knockout mouse) have found a new effector of G_i/o_ receptor signaling. The current amplitudes observed were small under physiological conditions, however the authors nonetheless showed that the channel has major effects on pacemaker firing in this cell type (consistent with previous work by the senior author showing that small sub-threshold currents can have large effects on firing). As midbrain dopaminergic neurons are involved in complex behavioral processes and the etiology of many neurological disorders, these findings could have far-reaching consequences. Furthermore, it is likely that these effects occur in at least some of the other neuron types throughout the CNS where G_i/o_ receptor signaling is a prominent inhibitory mechanism. The experiments themselves range from voltage clamp to current clamp to cell-attached recordings, and consistent with this group's reputation are expertly performed. The manuscript itself is well-written. However, the final contribution to the literature could be enhanced with a few straightforward additions.

In the G_i/o_ inhibition experiments, only single concentrations of drugs were used. Using GIRK channel activation in these neurons as a guide, 100 µM dopamine might be expected to be maximally effective while 10 µM baclofen might be expected to be somewhat less than maximal. Strikingly, 10 µM baclofen completely walloped the NALCN current (Figure 4A). This suggests that this effector may be more sensitive to GABAB receptor activation than GIRK channels, which would be extremely interesting and have implications for the physiological relevance of these findings. Full concentration-response curves with high n's are probably too much to ask. However, given that this manuscript is describing a new effector of G_i/o_ signaling in these heavily studied neurons, a rough concentration-response with maybe ~2 more concentrations of baclofen and dopamine would allow for an estimate of the EC50s, and would be a valuable addition for the literature record.

The D2 receptor desensitization finding should be further clarified. A previous report (Beckstead and Williams, 2007, J Neurosci, 27: 2074-2080) suggested that the percent desensitization of GIRK channel signaling produced by 100 µM dopamine in these same neurons is not different in slices from β-arrestin 2 KO mice. The current manuscript suggests a different result for NALCN inhibition, although the experiment may be a bit underpowered for high confidence (n=4 in the arrestin group). Oddly, the wild-type trace in Figure 2A shows zero dopamine-induced desensitization, but the summary data in panel 2D suggests that the smallest amount of desensitization observed in any single neuron in wild-type mice was around 15%. The authors should clarify why the trace doesn't match the summary data and maybe get a few more arrestin cells to enhance confidence in this conclusion.

*Reviewer #2:*

This study presents convincing evidence that NALCN-mediated currents are inhibited by G_i/o_ GPCR activation in nigral dopamine neurons. The manuscript will benefit from additional text clarifying methodological details and more general improvements to the clarity of the writing several of which are pointed out below. Additional pharmacological experiments that definitively show that dopamine is acting cell-autonomously to mediate the inhibition of NALCN current would be helpful.

While the experiments using NALCN cKO are convincing and key for the argument of NALCN-specific inhibition, the manuscript would benefit from a more detailed description of bath solution composition and control for osmolality/pH when solution changes are made. This is important for addressing a concern about leaky-patch clamp artifacts that the authors bring up in the third paragraph of the subsection “Determining the contribution of NALCN to sodium leak currents in SNc dopamine neurons”, which could be a real concern if change in osmolality/pH are altering the "leakiness" of the seal.

1) The authors should more clearly describe the composition of the bath solutions in the Results/Materials and methods section. How was osmolality and pH confirmed to be identical between different ACSF composition. In particular, what replaced calcium in the low calcium solution and was Na-bicarbonate (Na-HCO3) replaced in the low sodium solution and if so by what? If HEPES was used to replace Na-bicarbonate, was the solution bubbled with 100% oxygen instead of 95%Oxygen/5%CO_2_ to account for the different buffer? The authors should clarify that NMDG-chloride was used and that the pH did not change between solutions.

A potential confound could exist for the interpretation of the dopamine and baclofen experiments as cell-autonomous. The following pharmacological experiments would help make this more convincing and at the very least the authors should discuss this possibility. It would be sufficient to perform these additional experiments using cell-attached recordings of firing.

1) Since only picrotoxin was used to block GABA-A receptors and since dopamine application to slices could increase GABAergic neuron input onto the recorded cells, the effects could be mediated by GABA-B receptor suppression of NALCN current rather than direct D2 receptor mediated effects. Therefore, to make the major claim that D2 receptors are involved, the authors must show that dopamine can reduce NALCN current in the presence of GABA-B blocker (e.g. CGP35348). For the voltage clamp experiments, TTX was included which the authors should, at the very least, make a stronger point of arguing that it is unlikely that dopamine is increasing GABA spontaneous firing and thus releasing more GABA onto dopamine neurons in an action potential dependent manner. However, the pharmacological block of GABA-B receptors would directly rule this out this potential confound.

2) Similarly, but less concerning than point 1 above, the baclofen experiments should be performed in the presence of dopamine receptor blockers to rule out similar non-cell-autonomous effects.

3) Additionally, the experiments could benefit from D1 receptor antagonist (SCH23390), which would help prevent excitation of GABAergic neurons in the slice and rule out any effects of this receptor on the inhibition of NALCN current.

An even more convincing, but longer experiment would be to obtain the Drd2-Flox mice from Jackson Laboratory, eliminate the D2 receptor from dopamine neurons by crossing with DAT-cre, and show in slices that dopamine no longer can inhibit NALCN currents.

*Reviewer #3:*

The study by Philippart and Khaliq reports D2 and GABA-B receptor activation inhibits a tonic current carried by NALCN channels in substantia nigra pars compacta (SNc) dopamine neurons. The data are well-organized, very clearly illustrated, and intriguing. The only major concern is that it is difficult to determine if the experimental conditions employed to isolate NALCN exaggerates its magnitude and contribution to regulation of cell firing. A better understanding of the context in which this form of auto-regulation is important, physiologically, would be beneficial to all readers.

1) Since NALCN is permeable to Na^+^, K^+^, and Cs^+^ (Ren 2011), does the inclusion of Cs^+^ in and outside of the cell alter NALCN conductance measured in voltage-clamp? This may be tested by examining the effect of low calcium and NMDG on the tonic NALCN current using KMeSO_3_-based internal (as was done for current-clamp recordings).

2) Does Ba^2+^ affect NALCN? This becomes a concern if Ba^2+^ potentiates NALCN conductance; then the inhibitory effects of D2 receptor activation of GIRK and inhibition of NALCN are difficult to compare. This could be tested in voltage-clamp.

3) Does inhibition of NALCN slow dopamine neuron firing across a range of firing rates? The baseline firing rates after addition of BaCl_2_ and Ter-Q are much faster than in control conditions, presumably by block of other potassium channels by BaCl_2_. However, this could render the neurons more sensitive to very small hyperpolarizations and is not demonstration that D2 receptor inhibition of NALCN regulates firing in physiological conditions. Can this be tested using Ter-Q only (omitting Ba^2+^)? Alternatively, current injection may be employed to speed the firing rate of dopamine neurons and then puff DA, to test whether slowing of firing rate is more pronounced when the neurons are firing very quickly.

[Editors' note: further revisions were requested prior to acceptance, as described below.]

Thank you for resubmitting your work entitled "G_i/o_ protein-coupled receptors in dopamine neurons inhibit the sodium leak channel NALCN" for further consideration at *eLife*. Your revised article has been favorably evaluated by Gary Westbrook (Senior Editor) and the two of the original three reviewers. The reviewers are satisfied with your revisions but we would like you to address a few comments as below, before we can make final acceptance.

*Reviewer #2:*

The authors have completely addressed all of my original comments. They provide exciting and compelling evidence that dopamine actions via D2 receptors and baclofen actions via GABAB receptors can gate SNc neuron firing specifically via NALCN channels. The manuscript is clear and convincing. These data already motivate exciting future directions to study how NALCN contributes to the function of dopamine neurons.

*Reviewer #3:*

The revisions have greatly improved the manuscript and addressed the points raised in my review. The study is very interesting. There are only a few points remaining to consider.

1) The impact statement and Introduction (last paragraph) state that the NALCN mechanism functions synergistically with GIRK channels. "Synergistic" should be removed. This study does not address interactions between NALCN and GIRK to conclude that the effects are greater than the sum of the effect of D2/GABAB at each channel individually. The discussion regarding a potential synergistic effect is very informative and should remain.

2) It would be beneficial to the readers to include the data of NMDG-sensitive currents recorded with Cs-based internal compared with K-based internal, in Materials and methods or Results. Notably, presenting that GIRK and NALCN can be recorded with the same internal solution helps advance future studies.

---

## [Author Response]

Reviewer #1:[…] In the G_i/o_inhibition experiments, only single concentrations of drugs were used. Using GIRK channel activation in these neurons as a guide, 100 µM dopamine might be expected to be maximally effective while 10 µM baclofen might be expected to be somewhat less than maximal. Strikingly, 10 µM baclofen completely walloped the NALCN current (Figure 4A). This suggests that this effector may be more sensitive to GABAB receptor activation than GIRK channels, which would be extremely interesting and have implications for the physiological relevance of these findings. Full concentration-response curves with high n's are probably too much to ask. However, given that this manuscript is describing a new effector of G_i/o_signaling in these heavily studied neurons, a rough concentration-response with maybe ~2 more concentrations of baclofen and dopamine would allow for an estimate of the EC50s, and would be a valuable addition for the literature record.

This insightful suggestion has contributed significantly to the revised version of our manuscript. We thank the reviewer for this comment.

We performed new experiments to test the concentration dependence of NALCN inhibition by dopamine and baclofen. Concentration-response curves for dopamine inhibition of NALCN yielded an IC50 of 725 nM (Figure 2C), which is roughly comparable to published EC50 values for dopamine activation of GIRK channels in dopamine neurons (published dopamine EC50 values, 155 nM and 233 nM; Uchida et al., 2000; Kim et al., 1995).

Concentration-response curves for baclofen inhibition of NALCN yielded an IC50 of 267 nM (Figure 6A), which is more than an order of magnitude lower than published EC50 values for baclofen activation of GIRK channels in dopamine neurons (published baclofen EC50 values, 9.2 and 14.8 μM; Chan et al., 1998; Cruz et al., 2004). As the reviewer predicted, therefore, we find that NALCN channels are substantially more sensitive than GIRK2 channels to G protein signaling by GABA-B receptors.

This observation suggests that the effects of low baclofen concentrations may be mediated almost exclusively by inhibition of NALCN. Consistent with this idea, we found that spontaneous firing (measured in 300 nM Tertiapin-Q) was significantly reduced even by 300 nM concentration of baclofen (Figure 6C-E). These results have been added to the current version of the manuscript.

The D2 receptor desensitization finding should be further clarified. A previous report (Beckstead and Williams, 2007, J Neurosci, 27: 2074-2080) suggested that the percent desensitization of GIRK channel signaling produced by 100 µM dopamine in these same neurons is not different in slices from β-arrestin 2 KO mice. The current manuscript suggests a different result for NALCN inhibition, although the experiment may be a bit underpowered for high confidence (n=4 in the arrestin group). Oddly, the wild-type trace in Figure 2A shows zero dopamine-induced desensitization, but the summary data in panel 2D suggests that the smallest amount of desensitization observed in any single neuron in wild-type mice was around 15%. The authors should clarify why the trace doesn't match the summary data and maybe get a few more arrestin cells to enhance confidence in this conclusion.

The reviewer is correct to point this out. In the earlier version of this manuscript, we limited our analysis to cells that were exposed to dopamine for at least 70 seconds starting from the time of maximal inhibition. According to this criterion, we excluded the example trace shown in the first version of Figure 2A. However, we have since concluded that the slow timing of agonist onset during our bath application experiments will lead to a mixture of activation and desensitization, which will complicate examination of kinetics of receptor responses. Therefore, we have removed these data from the current manuscript.

Reviewer #2:[…] While the experiments using NALCN cKO are convincing and key for the argument of NALCN-specific inhibition, the manuscript would benefit from a more detailed description of bath solution composition and control for osmolality/pH when solution changes are made. This is important for addressing a concern about leaky-patch clamp artifacts that the authors bring up in the third paragraph of the subsection “Determining the contribution of NALCN to sodium leak currents in SNc dopamine neurons”, which could be a real concern if change in osmolality/pH are altering the "leakiness" of the seal.1) The authors should more clearly describe the composition of the bath solutions in the Results/Materials and methods section. How was osmolality and pH confirmed to be identical between different ACSF composition. In particular, what replaced calcium in the low calcium solution and was Na-bicarbonate (Na-HCO3) replaced in the low sodium solution and if so by what? If HEPES was used to replace Na-bicarbonate, was the solution bubbled with 100% oxygen instead of 95%Oxygen/5%CO_2_ to account for the different buffer? The authors should clarify that NMDG-chloride was used and that the pH did not change between solutions.

We thank the reviewer for raising this point. HEPES-based external solutions were not used in this study. Na-bicarbonate was not replaced in the low Na bath solution. In low calcium solutions, calcium was lowered without adjusting the concentration of magnesium. All solutions were bubbled with carbogen.

We have added the following to the manuscript:

“Low sodium solution experiments were performed using extracellular solutions in which 125 NaCl was substituted for N-methyl-D-glucamine (NMDG)-Cl. […] The osmolarity of both high and low sodium external solutions was typically in the range of 300-315 mOsM.”

A potential confound could exist for the interpretation of the dopamine and baclofen experiments as cell-autonomous. The following pharmacological experiments would help make this more convincing and at the very least the authors should discuss this possibility. It would be sufficient to perform these additional experiments using cell-attached recordings of firing.1) Since only picrotoxin was used to block GABA-A receptors and since dopamine application to slices could increase GABAergic neuron input onto the recorded cells, the effects could be mediated by GABA-B receptor suppression of NALCN current rather than direct D2 receptor mediated effects. Therefore, to make the major claim that D2 receptors are involved, the authors must show that dopamine can reduce NALCN current in the presence of GABA-B blocker (e.g. CGP35348). For the voltage clamp experiments, TTX was included which the authors should, at the very least, make a stronger point of arguing that it is unlikely that dopamine is increasing GABA spontaneous firing and thus releasing more GABA onto dopamine neurons in an action potential dependent manner. However, the pharmacological block of GABA-B receptors would directly rule this out this potential confound.

We performed new cell-attached experiments to test the effect of dopamine puff on spontaneous firing when cells were recorded in the presence of CGP 55845. We combined these experiments with the tertiapin-Q only experiments (as requested by reviewer 3; see below). In addition, we added SCH 39166 to inhibit D1 receptors (as requested here in reviewer 2 comment 3; see below). We reasoned that if the combination of these blockers altered the magnitude of the dopamine effect, then we would investigate each antagonist individually.

As shown in Figure 3—figure supplement 1, we found no difference in effect size of dopamine puff when cells were recorded with Ter-Q only, CGP 55845 and SCH 39166 added to the bath(p=0.54).

In addition, voltage-clamp experiments examining the concentration-dependence of dopamine were performed with CGP 55845, SCH 39166 and nomifensin added to the bath (Figure 3C).Similarly, dopamine inhibition of NALCN was still present suggesting that the effects result from agonist induced activation of intrinsically expressed D2 receptors.

2) Similarly, but less concerning than point 1 above, the baclofen experiments should be performed in the presence of dopamine receptor blockers to rule out similar non-cell-autonomous effects.

Again, we understand the concern but we believe that this scenario is unlikely for two main reasons. First, if the baclofen-mediated reduction in firing can be attributed to activation of dopamine receptors, then this would require bath application of 10 μM baclofen to increase dopamine neuron firing and/or somatodendritic release of dopamine. However, we are unaware of any published data showing increases in dopamine activity in response to high concentrations of baclofen. Secondly, the reviewer makes an excellent point that the voltage-clamp experiments using TTX argue strongly against action potential dependent effects originating from the surrounding circuit. Therefore, we maintain that the most parsimonious explanation for the inhibition of NALCN by baclofen is that it results from direct activation of GABA-B receptors present on the cell membrane.

3) Additionally, the experiments could benefit from D1 receptor antagonist (SCH23390), which would help prevent excitation of GABAergic neurons in the slice and rule out any effects of this receptor on the inhibition of NALCN current.An even more convincing, but longer experiment would be to obtain the Drd2-Flox mice from Jackson Laboratory, eliminate the D2 receptor from dopamine neurons by crossing with DAT-cre, and show in slices that dopamine no longer can inhibit NALCN currents.

As mentioned above, we performed new experiments to test the effect of puffing dopamine in the presence of the D1 receptor antagonist, SCH 39166. Adding 1 µM SCH 39166 to the bath had no effect on the dopamine-mediated inhibition of spontaneous firing recorded in cell-attached mode (Figure 3—figure supplement 1). In addition, experiments examining the concentration-dependence of dopamine were performed with CGP 55845 and SCH 39166 added to the bath (Figure 3C). Similarly, dopamine inhibition of NALCN was still present suggesting that the effects result from agonist induced activation of intrinsically expressed D2 receptors.

Reviewer #3:The study by Philippart and Khaliq reports D2 and GABA-B receptor activation inhibits a tonic current carried by NALCN channels in substantia nigra pars compacta (SNc) dopamine neurons. The data are well-organized, very clearly illustrated, and intriguing. The only major concern is that it is difficult to determine if the experimental conditions employed to isolate NALCN exaggerates its magnitude and contribution to regulation of cell firing. A better understanding of the context in which this form of auto-regulation is important, physiologically, would be beneficial to all readers.

Thank the reviewer for the positive assessment of our work. We have added text summarizing the physiological and behavioral importance of D2-receptor autoinhibition to the Introduction (Introduction, second paragraph).

1) Since NALCN is permeable to Na^+^, K^+^, and Cs^+^ (Ren 2011), does the inclusion of Cs^+^ in and outside of the cell alter NALCN conductance measured in voltage-clamp? This may be tested by examining the effect of low calcium and NMDG on the tonic NALCN current using KMeSO_3_-based internal (as was done for current-clamp recordings).

As suggested by the reviewer, we performed a set of experiments using KMeSO_3_-based internal solutions. We observed no difference in the amplitude of NMDG-sensitive currents recorded with Cs-based and K-based internal solutions (NMDG-sensitive current at -70 mV; Cs-based, 138 ± 18 pA, n=22; K-based, 107.9 ± 17 pA, n=8; p=0.36). These results show that the inclusion of Cs^+^ into the pipette does not significantly affect the NALCN current measured at -70 mV.

2) Does Ba^2+^ affect NALCN? This becomes a concern if Ba^2+^ potentiates NALCN conductance; then the inhibitory effects of D2 receptor activation of GIRK and inhibition of NALCN are difficult to compare. This could be tested in voltage-clamp.

We compared the amplitude of NALCN currents recorded in low Ca^2+^ and found no difference between currents recorded in the absence or presence of 100 μM Ba^2+^ (leak current at -70 mV; control, -192 ± 20 pA, n=22; Ba^2+^, -195.3 13.29, n=22; p=0.89). Therefore, 100 μM Ba2+ does not potentiate the NALCN current under our experimental conditions.

3) Does inhibition of NALCN slow dopamine neuron firing across a range of firing rates? The baseline firing rates after addition of BaCl_2_ and Ter-Q are much faster than in control conditions, presumably by block of other potassium channels by BaCl_2_. However, this could render the neurons more sensitive to very small hyperpolarizations and is not demonstration that D2 receptor inhibition of NALCN regulates firing in physiological conditions. Can this be tested using Ter-Q only (omitting Ba^2+^)? Alternatively, current injection may be employed to speed the firing rate of dopamine neurons and then puff DA, to test whether slowing of firing rate is more pronounced when the neurons are firing very quickly.

We tested for a correlation between the effect of dopamine inhibition and the rate of firing (recorded in 100 μM Ba^2+^ and 300 nM tertiapin). Our data shows little to no correlation between these variables (Pr=-0.298, n=17, p=0.24). This data has been added to the manuscript as Figure 3—figure supplement 2.

In addition, we performed new experiments to test the effect of dopamine puff in inhibiting cell-attached firing using only Ter-Q to block GIRKs. We found no difference in the effect of dopamine in cells that were treated with Ba^2+^ plus Ter-Q and those treated with Ter-Q alone (ANOVA p=0.57; Ter-Q vs. Ba^2+^ plus Ter-Q, p=0.54). We also tested the effect of a high concentration of Ba^2+^ (500 μM) alone which were similar to results in Ter-Q and Ter-Q plus Ba^2+^. This data has been added to the manuscript as Figure 3—figure supplement 1.

[Editors' note: further revisions were requested prior to acceptance, as described below.]

Reviewer #3:The revisions have greatly improved the manuscript and addressed the points raised in my review. The study is very interesting. There are only a few points remaining to consider.1) The impact statement and Introduction (last paragraph) state that the NALCN mechanism functions synergistically with GIRK channels. "Synergistic" should be removed. This study does not address interactions between NALCN and GIRK to conclude that the effects are greater than the sum of the effect of D2/GABAB at each channel individually. The discussion regarding a potential synergistic effect is very informative and should remain.

We agree, this word has been removed from the Introduction.

2) It would be beneficial to the readers to include the data of NMDG-sensitive currents recorded with Cs-based internal compared with K-based internal, in Materials and methods or Results. Notably, presenting that GIRK and NALCN can be recorded with the same internal solution helps advance future studies.

We added the summarized results for these experiments to the Results sections (subsection “Determining the contribution of NALCN to sodium leak currents in SNc dopamine neurons”).